# Immune and Microbial Signatures Associated with PD-1 Blockade Sensitivity in a Preclinical Model for HPV+ Oropharyngeal Cancer

**DOI:** 10.3390/cancers16112065

**Published:** 2024-05-30

**Authors:** Jennifer Díaz-Rivera, Michael A. Rodríguez-Rivera, Natalie M. Meléndez-Vázquez, Filipa Godoy-Vitorino, Stephanie M. Dorta-Estremera

**Affiliations:** 1Cancer Biology Division, Comprehensive Cancer Center, University of Puerto Rico, San Juan, PR 00936, USA; jennifer.diaz11@upr.edu (J.D.-R.); michael.rodriguez21@upr.edu (M.A.R.-R.); 2Microbiology and Medical Zoology Department, University of Puerto Rico Medical Sciences Campus, San Juan, PR 00936, USA; natalie.melendez2@upr.edu (N.M.M.-V.); filipa.godoy@upr.edu (F.G.-V.)

**Keywords:** oropharyngeal, HPV, PD-1 blockade, cancer, microbiota, diversity, biomarkers

## Abstract

**Simple Summary:**

The United States is suffering from an epidemic associated with infections with the Human Papilloma Virus (HPV) that are responsible for the development of head and neck squamous cell carcinoma (HNSCC). Our study aims to characterize immune profiles, and the oral microbiota in oropharyngeal tumors associated with anti-PD-1 treatment response in a preclinical murine model for HPV^+^ oropharyngeal cancer. We confirmed 16 immune markers and specific bacteria in the oral cavity that differentiate between treatment responders and non-responders, which may be used as potential biomarkers for PD-1 blockade response.

**Abstract:**

The United States is suffering from an epidemic associated with high-risk strains of the Human Papillomavirus (HPV) predominantly responsible for the development of head and neck squamous cell carcinoma (HNSCC). Treatment with immune checkpoint inhibitors targeting programmed death 1 (PD-1) or its ligand PD-L1 has shown poor efficacy in HNSCC patients, observing only a 20–30% response. Therefore, biological marker identification associated with PD-1 blockade response is important to improve prognosis and define novel therapeutics for HNSCC patients. Therapy response was associated with increased frequencies of activated CD27^+^T cells, activated CD79a^+^ B cells, antigen-presenting CD74^+^ dendritic and B cells, and PD-L1^+^ and PD-L2^+^ myeloid-derived suppressor cells (MDSCs). The oral microbiota composition differed significantly in mice bearing tongue tumors and treated with anti-PD-1. A higher abundance of *Allobaculum*, *Blautia*, *Faecalibacterium*, *Dorea,* or *Roseburia* was associated with response to the therapy. However, an increase in *Enterococcus* was attributed to tongue tumor-bearing non-responding mice. Our findings indicate that differences in immune phenotypes, protein expression, and bacterial abundance occur as mice develop tongue tumors and are treated with anti-PD-1. These results may have a clinical impact as specific bacteria and immune phenotype could serve as biomarkers for treatment response in HNSCC.

## 1. Introduction

Head and neck squamous cell carcinoma (HNSCC) is the sixth most common cancer, with an estimated 2020 worldwide burden of over 890,000 new cases, out of which 54,000 cases were reported in the U.S. [1,2]. The incidence of HNSCC is increasing over time and is estimated to increase by 30% by 2030 [3]. The incidence of HNSCC varies across different ethnic groups in the United States. It is slightly higher among black individuals, while in Puerto Ricans, the incidence rate is 2.5 times higher when compared to other Hispanics living in the U.S. [4,5,6]. Oropharyngeal squamous cell carcinoma (OPSCC), a subset of HNSCC, is a malignancy that develops within the posterior third of the tongue, soft palate, tonsils, and posterior pharyngeal wall [7]. One of the most prevalent risk factors for OPSCC is human papillomavirus (HPV) infection, where 70% of oropharyngeal cancers are associated with HPV [8]. Current treatment for OPSCC consists of surgery, chemotherapy, radiation therapy, and, more recently, immune checkpoint inhibitors (ICIs). One of the most common ICIs targets the lymphocyte immune inhibitory receptor PD-1 (programmed death 1) with monoclonal antibodies (anti-PD-1), thus competing with its ligands programmed death-ligand 1 (PD-L1) and PD-L2, expressed by tumors and immunosuppressive cells, preventing cell anergy and promoting tumor destruction [9,10,11]. Although this therapy has seen success in multiple cancers, such as melanoma and cervical cancer, it has displayed very poor efficacy in treating OPSCC, with about 20–30% of therapy response and full recovery [9,10,11,12]. The poor responsiveness to anti-PD-1 therapy in HPV^+^ OPSCC is multifactorial and requires more studies to elucidate it.

Anti-PD-1 therapy has been proven to increase the infiltration of anti-tumor immune cells, such as CD8^+^ T cells, thus overcoming the immunosuppressive profile associated with tumors [13]. Nevertheless, studies have identified that in OPSCC, the tumor microenvironment (TME) may display increased infiltration of immune cells with pro-tumoral and anti-inflammatory characteristics, such as myeloid-derived suppressor cells (MDSCs) and T regulatory cells [14,15,16].

In addition, the interaction of immune cells with the microbiota, especially in mucosal tissues, affects immunological function and, thus, tumor development. The microbiota consists of symbiotic microbial cells that can regulate the immune system, protect against pathogens, and promote anti-tumor immune responses. These are important to counteract tumor growth and promote treatment efficacy [17,18]. Hence, studying the immune-microbial interplay in cancer is extremely important [19]. The gut microbiome has been associated with influencing cancer progression and the effectiveness of immune checkpoint inhibitors (ICIs). This occurs through the microbiome’s role in directly stimulating immune responses and producing metabolites that may lead to the destruction of cancer cells or alternatively support their survival. Previous studies in non-small cell lung cancer revealed that patients treated with PD-1 blockade showed higher microbial diversity and increased abundance of CD8^+^ T cells and NK cells [20]. Specifically, research indicates that certain gut bacteria, such as *Bifidobacterium*, enhance the immune response to tumors by increasing the production of interferon-gamma (IFN-γ) and promoting T cell activation and co-stimulation [21]. In preclinical mouse models, oral administration of *Bifidobacterium* has been shown to improve the effectiveness of anti-PD-L1 therapy [22].

While most studies have focused on the gut microbiota, recent studies have extended beyond the gut to include the oral cavity, which is known to harbor approximately 700 bacterial species and may be involved in tumor development and ICI response. Members of the human oral microbiota involved in maintaining oral health include *Streptococcus* species, notably abundant across different oral habitats, along with genera such as *Neisseria*, *Prevotella*, and *Haemophilus* [23]. In mice, *Porphyromonas gingivalis* and *Fusobacterium nucleatum* can induce tumor development. Additionally, in HNSCC, there has been a significant association with Lactobacillus and tumor progression [24,25,26]. In tongue cancer, researchers have found an enrichment of *Fusobacterium*, and although not associated with immune cell infiltration, it was associated with increased PD-L1 mRNA expression. Indeed, there is a relationship between the presence of *Fusobacterium* and increased PD-L1 expression in oral tongue squamous cell carcinoma, suggesting the bacterium’s capacity to trigger PD-L1 expression and thus being a biomarker that possibly influences the immune environment within head and neck cancer [27].

Taken together, these studies indicate that the immune and oral microbial composition may affect immune checkpoint blockade efficacy in oropharyngeal carcinoma. However, a comprehensive study that integrates immune analysis of lymphoid organs, tumors, and blood, as well as microbiota analysis of tumors, is important to identify novel prognostic markers for immune checkpoint blockade efficacy. This study provides an in-depth characterization of the immune profiles in tumors, tumor-draining lymph nodes and blood, and the oral microbiota in HPV+ oropharyngeal tumors and their association to anti-PD-1 therapy response, using a preclinical murine model for HPV^+^ oropharyngeal cancer.

## 2. Materials and Methods

### 2.1. Animals

Male C57BL/6 mice (5–10 weeks) were purchased from the Jackson Laboratory, maintained in a pathogen-free environment, and caged in groups of 5. Animals were anesthetized with isoflurane for tumor inoculations. Anesthetized mice were euthanized via cervical dislocation. All animal studies were pre-approved and carried out in accordance with the University of Puerto Rico-Medical Sciences Campus Institutional Animal Care and Use Committee (IACUC) protocol #A630122.

### 2.2. Cell Line

Mouse tonsil epithelial cells expressing HPV-16 E6 and E7, as well as H-Ras (mEER), were kindly donated by Dr. Paola Vermeer (University of South Dakota, Vermillion, SD, USA). These cells were maintained in complete media as previously described and sub-cultured at 80% confluence the day before tumor induction in mice [28].

### 2.3. Reagents

The tumor-infiltrating leukocytes (TILs) and tumor-draining lymph nodes were analyzed by multi-parametric flow cytometry. The following antibodies from Biolegend (San Diego, CA, USA) were used: BV650 anti-CD3 (clone 17A2), BV421 anti-CD11b (clone M1/70), FITC anti-CD74 (clone IN1/CD74), anti-mouse CD16/32 (2.4G2, mouse Fc-block), BV711 anti-Gr1(clone RB6-8C5), BV786 anti-PDL-1 (clone 10F.9G2), PE-CF594 anti-CD4, PECy7 anti-CD273 (clone TY25), Pacific Blue anti-CD44 (clone NIM-R8), BV605 anti-CD11c (clone N418), BV605 anti-CD138 (clone 281-2), BV650 anti-CD19 (clone 6D5), BV650 anti-CD27 (clone LG.3A10), BV711 anti-CD4 (clone GK1.5), BV786 anti-CD45R (clone RA3-6B2), FITC anti-CD3 (clone 17A2), PE anti-CD79A (clone F11-172), PE-CF594 anti-CD335 (NKp46) (clone 29A1.4), PerCP-Cy5.5 anti-CD8 (clone 53-6.7). For in vivo administration, we used 250 µg dose per mouse of anti-PD-1—clone RMP 1-14 (Leinco Technologies, Fenton, MO, USA).

### 2.4. In Vivo Tumor Challenge

Mice were implanted 5 × 10^4^ mEER cells resuspended in 50 µL of PBS into the base of the tongue and monitored by visual observations, confirming tumor establishment on day 10. Mice received intraperitoneal injections of anti-PD-1 antibody (250 µg/mouse) on day 10, and two additional doses were administered at 3-day intervals (days 13 and 16). Animals were divided according to therapy response, determined by the presence or absence of the tumor after 25 days. Mice with tongue tumors were euthanized when they lost ≥20% of their initial weight or following endpoint criteria already approved by the IACUC. A total of 42 mice were used for immune phenotyping analyses, and 27 were used for microbiota analysis.

### 2.5. Flow Cytometry and Data Analyses

For the characterization of leukocytes, mice were euthanized on day 29 after the tumor challenge. As previously described, tongues and primary tongue tumors were collected and digested [29]. The cervical lymph nodes were collected and identified as tumor-draining lymph nodes (tdLNs). Purified leukocytes were stained for multi-parametric flow cytometry analysis with a 10-color antibody panel. Two panels were used per sample. For intracellular staining of CD79a, cells were blocked with mouse Fc block, stained with surface markers, fixed and permeabilized with the BD Cytofix buffer (Biotechne, Minneapolis, MN, USA) followed by staining for intracellular markers with BD Perm Buffer. Samples were run in a 2-laser FACS Celesta flow cytometer (BD Biosciences, San Jose, CA, USA) and analyzed using FlowJo version 10.9.0 (FlowJo LLC, Ashland, OR, USA). All flow cytometry data statistics were calculated using GraphPad Prism version 10 (Boston, MA, USA). For the flow cytometry analysis, a Shapiro–Wilk test was used to determine the normality of the distributions. Statistical significance of the normal distributions was assessed with a two-way ANOVA, while distributions that did not display normality were analyzed with a Kruskal–Wallis (KW) test. *p* values less than 0.05 were considered significant (* *p* < 0.05). All data were pooled from two experiments (*n* = 42), except for the NKp46 immune expression data (*n* = 25). The samples were divided into groups including naïve (*n* = 10), animals with tumors who did not receive treatment (*n* = 8), those under treatment who responded to the therapy (*n* = 16), and non-responders (*n* = 8).

To observe the relationships in the data between the different mouse groups, we compared cell type frequencies and protein marker data extracted from the flow cytometry analysis. These data were input into GraphPad Prism version 10 (Boston, MA, USA) to generate a principal component analysis (PCA). The data were standardized and scaled to have a mean of 0 and a standard deviation of 1, and principal components (PCs) were selected based on parallel analysis.

### 2.6. Oral Microbiota Experiments and Data Analysis

#### Genomic DNA Extraction, Amplification, and Sequencing

Genomic DNA (gDNA) extraction was performed on the tongue (*n* = 15) and tumor samples (*n* = 12) using the QIAGEN DNAeasy Powersoil Pro Kit (QIAGEN, Hilden, Germany) following the manufacturer’s protocol. DNA was quantified using the Qubit dsDNA HS (High Sensitivity) Assay (ranging from 5 to 100 ng/µL) (Thermo Fisher Scientific, Waltham, MA, USA). The resulting gDNA was stored at −20 °C until amplification and sequencing.

Amplification was performed with the universal bacterial primers 515F and 806R following the Earth Microbiome Standard Protocols (https://earthmicrobiome.org/protocols-and-standards/) (accessed on 16 March 2024). The hypervariable V4 region of the 16S ribosomal RNA gene was sequenced following a paired-end community using 2 × 250 bp protocol with Illumina Miseq.

### 2.7. Read Processing, Clustering and Abundance Table

Demultiplexed bacterial reads from two different experiments were deposited and pre-processed with a Phred offset of 30 in the open-source platform QIITA [30]. A meta-analysis process was performed to join both individual projects. Trimming was performed at 250 bp, and closed-reference clustering was executed using the reference database SILVA with operational taxonomic units (OTUs) with a 97% identity [31,32]. The abundance table was downloaded, and further quality processing in QIIME2 included the removal of chloroplasts and mitochondria. Microbial analysis was continued in QIIME2 with a sampling depth of 1000. Data are available in QIITA under the study ID: 14411 and have been uploaded to the European Nucleotide Archive (ENA) with accession number PRJEB74643 ERP159289.

### 2.8. Microbial Ecological Analysis: Beta Diversity Distances and Alpha Diversity Estimates

Beta diversity was computed from a distance matrix with the Bray–Curtis distance (a measure of dissimilarity), and differences between microbial community composition of groups were visualized with Principal Component Analyses and visualized with QIIME2 view (https://view.qiime2.org/) (accessed on 16 March 2024). Statistical tests were permutational multivariate analysis of variance (PERMANOVA) employed using 999 permutations.

Alpha diversity was determined through the Shannon Index, considering both the species richness and distribution within our samples. Alpha rarefaction curves were built using QIIME2 and visualized with QIIME2 view (https://view.qiime2.org/) (accessed on 16 March 2024). Kruskal–Wallis pairwise *p*-values were obtained from QIIME2.

The compositional taxonomic barplot at the phylum level was generated and visualized with QIIME2. For the genus level profile, obtained from MicrobiomeAnalyst (33), we used additional filtering parameters, including the following: (1) minimum count of 4, (2) 40% prevalence in all samples, (3) 10% taxa removal based on inter-quantile range, and (4) total sum scaling [33].

### 2.9. Putative Bacterial Biomarker Analysis Using Complementary Methods

We performed putative biomarker analyses using two complementary methods. First, a linear discriminant analysis (LDA) effect size (LEfSe) was used, which is an algorithm that identifies the taxa features most likely to explain differences between groups [34]. Indeed, this method employs a non-parametric factorial Kruskal–Wallis (KW) sum-rank test to identify taxa that are differentially abundant among the groups of mice responders, followed by a linear discriminant analysis to determine the effect size of each identified taxa per group and rank them accordingly. We uploaded a feature table in MicrobiomeAnalyst (33) using filtering parameters including (1) minimum count of 4, (2) 40% prevalence in all samples, (3) 10% taxa removal based on inter-quantile range, and (4) total sum scaling. The LEfSe analyses resulted in bar plots and respective corrected FDR *p*-values with a cutoff of 0.05. The second method employs the Random Forest (RF) classification algorithm (a supervised machine learning tool) using the statistical analysis module in MicrobiomeAnalyst (33). The parameters used were a taxonomy level of genus, 5000 trees, 10 predictors, and the randomness setting “on”. The output file was a dot plot based on the mean decrease accuracy (y-axis) identifying the predicted taxa to explain each class (mice response group). We also plot the error rate for the classifications per animal group. The out-of-bag (OOB) error corresponds to the mean prediction error.

## 3. Results

### 3.1. Principal Component Analysis of Immunological Profiles Associated with PD-1 Blockade Response

To characterize the immune and molecular patterns associated with anti-PD-1 therapy response in HPV^+^ oropharyngeal cancer, we performed a flow cytometry analysis of lymphocytes isolated from tumor-draining lymph nodes (tdLNs) from tongue tumor-bearing mice treated with PD-1 blockade and those untreated. The analysis of tumor-infiltrating lymphocytes (TILs) and lymphocytes from tdLN provides information regarding antigen presentation, lymphocyte activation, and expansion at secondary lymphoid organs and the tumor in relation to tumor growth and immunotherapy response. First, to observe the relationships between the immunological data, we performed a principal component analysis (PCA) of the percent expression of 26 variables that included cell type frequencies and protein markers obtained from tdLNs, analyzing the mice groups: naïve (no tumor challenge and no anti-PD-1 treatment), no treatment (tumor challenge without anti-PD-1 treatment), non–responder (no tumor clearance after treatment), and responder (tumor clearance after treatment) mice. To observe these relationships, we produced a PC scores plot to project the original flow cytometry data into a two–dimensional space and stratify the data to discover underlying relationships between the mice groups. In our plot, the naïve group corresponds to the uppermost right cluster, the no–treatment group corresponds to the uppermost left cluster, the responder group corresponds to the lower right cluster, and the non-responder group is located between the middle and lower left clusters (Figure 1). The clustering of the data points observed in the graph indicates that the data analyzed in this study can correctly differentiate between each group. Thus, the mice in our study express specific immune and molecular patterns that best describe their response to the tumor challenge (if subjected) and the therapy (if treated). Moreover, by observing the overlap between the data points of each mouse group, we can infer that naïve and responder mice, as well as the no–treatment and non–responder mice, display high similarities with each other.

### 3.2. Leukocyte Frequencies Associated with PD-1 Blockade Response

We wanted to identify the specific immune signatures associated with each mouse group; therefore, we compared the frequencies of CD4^+^ and CD8^+^ T cells, NKp46^+^ natural killer (NK) cells, B220^+^ B cells, CD138^+^B220int plasmablasts, CD138^+^B220^−^ plasma cells, CD11b^+^ myeloid cells, CD11c^+^ dendritic cells, and Gr1^+^ MDSCs following our gating strategy as depicted on Appendix A between each mouse group. By using parts-of-whole pie charts, we illustrate the differential proportions of immune cell types between the different mouse groups (Figure 2A). We identified that mice therapy responders displayed significantly increased frequency of anti-tumor immune cells, such as CD4^+^ and CD8^+^ T cells, antigen-presenting dendritic cells, myeloid cells, and a trend for increased frequency for NK cells (Figure 2B). In contrast, non-responder mice displayed a significant increase in the frequency of B220^+^ B cells, as was described by our previous publication [35], and a trend for increased frequency for antibody-producing plasma cells (CD138^+^B220^−^) (Figure 2B). To observe the dimensional and spatial clustering of the immune cell subsets in our flow cytometry data, we also produced representative t-distributed Stochastic Neighbor Embedding (t-SNE) maps for our different mouse groups depicting T, NK, and B cell subsets (Appendix A), as well as myeloid and dendritic cells, MDSCs, and B cells (Appendix A). We observe similarities between t-SNE maps from naïve and responder mice, as well as non-treated and non-responder mice.

Next, we wanted to quantify the immune cell frequencies from lymphocytes isolated from primary HPV^+^ tumors (no treatment and non-responder mice) and tongues from mice with no tumor (naïve and responder mice). To achieve this, we performed parts-of-whole pie charts depicting the percentages of all cells examined in the flow cytometry analysis (Figure 2C). Our results depict that the immunotherapy responders had an increased percentage of overall CD8^+^ T cells, CD138^+^B220int plasmablasts, and dendritic cells but a decreased percentage of B cells and CD11b^+^ myeloid cells compared to naïve mice. Tumors from therapy non-responders displayed an increased overall percentage of CD138^+^B220int plasmablasts, CD138^+^B220^−^ plasma cells, CD11b^+^ myeloid cells, CD11c^+^ dendritic cells, and Gr1^+^ MDSCs. Still, non-responders showed decreased frequencies of CD8^+^ T cells and B cells compared to the untreated tumors. Thus, these results suggest that anti-PD-1 treatment changes lymphocyte infiltration into tumors. After comparing the immune phenotype percentages depicting therapy response (responder vs. non-responder), we identified that immunotherapy responders displayed an increased percentage of CD8^+^ T cells, B cells, and CD138^+^B220int plasmablasts but a lower overall percentage of CD138^+^B220^−^ plasma cells, CD11b^+^ myeloid cells, CD11c^+^ dendritic cells, and Gr1^+^ MDSCs (Figure 2C). These data demonstrate that tdLNs of responder mice are dominated by increased frequencies of CD4^+^ and CD8^+^ T cells, myeloid, dendritic, and natural killer cells. In contrast, nonresponders have increased frequencies of B cells.

### 3.3. Immune Phenotypes Associated with PD-1 Blockade Response

By using this preclinical model, we also characterized known biomarker proteins already associated with cancer therapy response in other cancer types. We performed a flow cytometry analysis identifying the expression of the activation proteins CD27 and CD79a, antigen-processing associated protein CD74, and inhibitory molecules PD-L1 and PD-L2 in different immune cell types between the four groups in tdLNs (Figure 3A–D). First, we analyzed the mean fluorescence intensity (MFI) of CD27 in CD4^+^ and CD8^+^ T cells, as well as in NK cells. We observed that therapy responders displayed a significantly increased expression of CD27 in both CD4^+^ and CD8^+^ T cells compared to therapy non-responders. Still, there were no significant changes in the expression of CD27 in NK cells between these groups (Figure 3A). We also analyzed the mean fluorescence intensity (MFI) of CD27 and the activation-associated protein CD79a, expressed only on B cells. No significant differences in CD27 expression were observed, but the protein CD79a was significantly more expressed in therapy responders and the no-treatment group when compared to non-responders (Figure 3B).

Next, we described the expression of the immunomodulatory molecules PD-L1 and PD-L2, where we identified that expression of both molecules was significantly increased in MDSCs from immunotherapy responders. However, PD-L1 displayed a trend for increased expression in B cells on non-responders when compared to the other groups. (Figure 3C). Expression analysis of the antigen–processing associated protein CD74 identified that therapy responder mice had a significantly increased expression of this protein in dendritic cells, with a similar trend for B cells (Figure 3D).

Lastly, we described the expression of CD27, CD79a, CD74, PD-L1, and PD-L2 in lymphocytes isolated from tongues (naïve and responder mice) and from primary tumors (no treatment and non-responder mice) (Appendix A). Although we did not identify statistically significant differences when analyzing CD27, CD44, and CD79a, we detected a trend for increased expression of PD-L1 and PD-L2 in MDSCs and PD-L1 on B cells from responder mice compared to non-responders (Appendix A). We also identified a significantly increased expression of the protein CD74 in dendritic cells from responder mice but decreased expression in B cells when compared to non-responders (Appendix A). We attribute the lack of significance in the expression patterns of some of these proteins to a small sample size since tumors had to be pooled to obtain a sufficient number of lymphocytes for the analysis.

### 3.4. Immune Profiles in Peripheral Blood Associated with PD-1 Blockade Response

To determine whether the immunological changes identified in lymphoid organs and tumors after treatment are also present in peripheral blood, we performed flow cytometric analyses in the whole blood of a small group of mice (*n* = 11) before (day 9), during (day 16), and after therapy was administered (days 23 or 29) (Appendix A). We identified that as the tumor develops, frequencies of neutrophils and plasma cells increase; in contrast, both immune cells declined upon anti-PD-1 treatment in responders but not in non–responders (Appendix A). The frequencies of CD19^+^ B cells decline upon tumor development; however, only therapy responders displayed a trend for B cell recovery after therapy was administered (Appendix A). Interestingly, before therapy, non–responder mice showed the highest expression of PD-L1 in neutrophils compared to the other groups, which declined upon treatment. In contrast, responder mice showed lower PD-L1 expression on neutrophils and remained constant upon treatment **(**Appendix A). Lastly, PD-L2 expression on B cells was higher in non–responders before and after therapy compared to untreated and responder mice (Appendix A). All other parameters analyzed in tdLNs and tumors were also analyzed in blood; however, no significant differences were observed. In summary, these results may identify PD-L1 on neutrophils and B cells and PD-L2 on B cells as potential biomarkers in blood that predict responsiveness to PD-1 blockade.

### 3.5. Changes in the Microbiota Associated with PD-1 Blockade Response

For oral microbiota characterization, our experimental design consisted of four experimental groups: (1) naïve (*n* = 9), (2) tumor–implanted mice with no treatment (*n* = 7), (3) anti-PD-1 responders (*n* = 6), and (4) anti-PD-1 non-responders (*n* = 5). Tongue and tumor sample collection was performed on 27 animals at a 30-day timepoint where 16S *rRNA amplicon gene* sequencing targeting the hypervariable V4 region was conducted. Quality assessment recovered an average of 22,861 good-quality reads after chloroplast and mitochondria filtration. The reads produced an average of 410 OTUs from all 27 samples, and the different categories are shown in Appendix A. The higher number of reads was exhibited by the no–treatment group, which were mice bearing tongue tumors but did not receive anti-PD-1 therapy (Appendix A). To control for bias, we used a sampling depth of 1000 reads per sample for further downstream microbial analyses.

The first microbial assessment quantified bacterial diversity among the experimental groups (Figure 4). The alpha diversity metric, using the Shannon Index, only showed significant differences between the naïve animals and the non-responders (KW *p*-value = 0.013) (Figure 4A, Appendix A). When studying microbial structure and composition through a beta diversity plot, we found significant differences between the naïve group with therapy responders (PERMANOVA *p*-value = 0.001) and non-responders (PERMANOVA *p*-value = 0.002) (Figure 4B, Appendix A). Also, the tumor-bearing mice that had no treatment showed structural differences with anti-PD-1 non-responding mice (PERMANOVA *p*-value = 0.016) and immunotherapy responders (PERMANOVA *p*-value = 0.003) (Figure 4B, Appendix A). Tumor-bearing animals that were not treated with anti-PD-1 remained with a similar microbiota to the naïve group (PERMANOVA *p*-value = 0.536) (Figure 4B). When comparing responders and non-responders, we also observed changes in the composition (PERMANOVA *p*-value = 0.017) (Figure 4B). Lastly, the beta diversity analysis shows that anti-PD-1 treatment alters the tumor microbial composition.

When looking at the taxonomic profiles between groups, we found 35 different phyla. Proteobacteria was reduced in the anti-PD-1 group compared to tumor-bearing mice with no treatment (Figure 4C). Immunotherapy responders had Bacteroidetes as the most dominant phylum, followed by Proteobacteria. However, the non-responders had a higher relative abundance of Firmicutes when compared to the other groups of animals (Figure 4C). Furthermore, with the 40% prevalence filtering, we detected a total of 17 genera, of which *Pseudomonas*, *Streptococcus*, *Blautia*, and *Acinetobacter* were the most abundant in the responders group when compared to the non-responders (Figure 4D). *Enterococcus* abundance was higher in the non-responders, while animals that received no treatment had a higher abundance of *Staphylococcus* (Figure 4D). Naïve animals presented a general abundance profile similar to the non-treatment group (Figure 4D).

The genus-level putative biomarker analysis employing the linear discriminant analysis effect size (LEfSe) selected 26 taxa statistically associated with the animal treatment class. Known probiotic and anti-inflammatory taxa present in tongue and tongue tumor samples of responders included *Faecalibacterium* FDR *p* = 0.0145, *Roseburia* FDR *p* = 0.0109, *Colinsella* FDR *p* = 0.0083, *Pseudobutyrivibrio* FDR *p* = 0.0083, or *Dorea* FDR *p* = 0.0078 (identified with LEfSE) (Figure 5A), protective gut taxa [36,37]. Most of these taxa were also detected by Random Forest (despite the error rate of the model being moderately high-57%) (Figure 5B,C) [36,37]. Also, *Blautia* FDR *p* = 0.0109) and *Allobaculum* (the most well-predicted taxa for responders in the Random Forest model) were higher in anti-PD-1 responding mice, while *Staphylococcus* (FDR *p* = 0.0089) lowered its abundance with the treatment and were dominant in non-treated animals (Figure 5). Additionally, *Enterococcus* (FDR *p* = 0.0414) detected in RF was associated with the non-responder group (Figure 5A,B). These results suggest that microbial diversity and composition differ between the anti-PD-1 responders and non–responders.

## 4. Discussion

The success of immunotherapies, such as PD-1 blockade, as cancer treatments are mostly attributed to the increased tumor recognition and cytotoxicity by CD8^+^ T cells in the tumor microenvironment [38]. It is undeniable that other immune cells, as well as microbial cells, interact with and can eliminate cancer cells or promote tumor growth via direct or indirect mechanisms [39]. Therefore, a better understanding of the tumor microenvironment, including immune responses and microbiota composition, could elucidate the mechanisms involved in PD-1 blockade resistance and thus improve its efficacy. Our study has identified 16 immune markers and specific bacteria in the tongue or tumor that differentiate between PD-1 blockade responders and non-responders in oropharyngeal cancer, which may be used as potential biomarkers for PD-1 blockade response in HNSCC patients.

As expected, therapy response was associated with cytotoxic immune cells such as CD8^+^ T cells in tumor and tdLNs, and NK cells in tdLNs, both of which are known for their potent anti-tumor immunity and have also been described to display memory-like responses once matured [40,41]. Also, therapy response increased CD4^+^ T cell frequencies, which have recently been shown to play a more significant role in tumor immunity than previously thought, not only by secreting cytokines such as IFNγ and TNFα to promote tumor cell apoptosis and senescence but also by displaying cytotoxic functions through the secretion of granzymes [42,43,44]. T cells from tdLNs of therapy responders displayed activated phenotypes due to increased expression of the costimulatory molecule CD27, which has been associated with better survival of HNSCC patients during chemotherapy [45]. Although naïve T cells express CD27, this molecule is upregulated on central memory T cells. It is known to mediate strong cytokine production and cytotoxicity to tumors; thus, it may serve as a marker for better therapy response [46,47]. The activated profile of T cells on therapy responders correlates with an increased expression of CD74 on dendritic cells, playing a pivotal role in antigen presentation through MHC class II to T cells [48].

We previously described an increased frequency of CD3^+^ T lymphocytes and CD19^+^ B cells in anti-PD-1 responding tongue tumors, whereas, in tdLNs, only T cell frequencies but not B cells were higher in therapy–responding mice compared to non-responders [35]. B cells from these secondary lymphoid tissues on therapy non–responders seem to possess an immunomodulatory profile observed by the highest PD-L1^+^ B cell frequencies in this group. Also, increased B cell frequencies in non-responder mice were accompanied by a decreased abundance of T and NK cells, which showed a less activated profile [49]. This supports previous studies that have observed the accumulation of regulatory-like B cells expressing immunosuppressive molecules, such as PD-L1 and PD-L2, known to cause T cell anergy and are associated with poor patient prognosis in tdLNs [49,50]. In contrast, our data indicate that the anti-PD-1 therapy response promotes B cells to express high levels of CD79a (Igα), a component of the B cell receptor complex heterodimer needed for signal initiation and transduction. Also, CD74 was highly expressed in B cells from anti-PD-1 responders [9,51,52,53]. The increased expression of CD79a and CD74 on B cells in tdLNs suggests that a subset of B cells acts as antigen-presenting cells in these lymphoid organs, promoting the activation of T cells. Importantly, CD79a and CD74 have been associated with anti-tumor response in OPSCC and in melanoma patients, respectively [52,54]. Interestingly, we observed increased tumor infiltration of plasmablasts in mice that responded to the therapy and increased infiltration of plasma cells in mice that did not respond to the therapy. The role of antibody-producing short-lived plasmablasts and long-lived plasma cells in tumor immunity has remained inconclusive. Some reports associate antibodies with anti-tumor responses by their ability to promote antibody-dependent cell-mediated cytotoxicity, phagocytosis, and antigen presentation. In contrast, other reports describe regulatory effects through the constitutive production of IL-10 by plasma cells [55,56,57]. Therefore, more analyses are needed to determine the specific roles of plasmablasts and plasma cells in HPV+ oropharyngeal cancer.

On the other hand, the infiltration of MDSCs was characteristic of tumors from non-responders. The role of MDSCs in the tumor microenvironment has been thoroughly explored, and they are heavily associated with immunosuppression [58]. In contrast, therapy responders had increased expression of the immunosuppressive molecules PD-L1 and PD-L2 by myeloid and dendritic cells in tdLNs and MDSCs in tumors. Although PD-L1 and PD-L2 are mostly associated with tumor progression, higher levels of these molecules are associated with better anti-PD-1 therapy response in several cancers [59,60].

The microbial composition also impacts immune responses toward tumors, thus, immune checkpoint blockade efficacy. Our data showed microbial associations with response to treatment, highlighting differences in the composition and diversity of these oral communities and including butyrate-producing taxa typically found in the gut. We determined that tumor-bearing animals that did not respond to the therapy showed a decrease in the Bacteroidetes phylum, an integral component of the normal oral flora, and an increase in Firmicutes and *Enterococcus*. Our results suggest a higher Firmicutes to Bacteroidetes ratio in non-responding mice, which has been associated with impaired homeostasis in the gut [61]. Specifically, an increase in Firmicutes and a decrease in Bacteroidetes are more associated with inflammatory conditions such as inflammatory bowel disease, obesity, and caries in the oral cavity [62]. Also, the phylum Firmicutes is more abundant in the oral cavity of oropharyngeal carcinoma patients, and changes in their levels may be more associated with pre-cancerous lesions [63,64]. This phylum is associated with Th17 cells that promote neutrophil recruitment and activation [65,66,67]. This is similar to our results, which demonstrate increased frequencies of MDSCs, known to share similar markers to neutrophils, in non–responding mice. The abundance of Bacteroidetes has been associated with the induction of IFN-γ producing T cells to protect the host from bacterial infections [68]. In addition, this phylum is highly recognized for its ability to produce short-chain fatty acids known to promote the migration of neutrophils to areas of inflammation and the differentiation of B cells into plasma cells [69,70,71]. On the other hand, *Enterococcus* has been associated with lesions in the oral cavity, such as periodontitis, and in promoting colorectal cancer [72,73]. Previous studies found that *Enterococcus* possesses different strategies to suppress, evade, or inactivate innate and adaptive immune responses, which may explain its high abundance in animals that did not respond to the treatment [74]. Specifically, *Enterococcus* can reside within macrophages and neutrophils, stimulating the secretion of cytokines such as IL-6 and IL-10, which are associated with tumor development and immunosuppression [75,76,77].

Other bacteria were detected as possible biomarkers for treatment response, including *Dorea*, *Roseburia*, *Allobaculum,* and *Blautia,* which were distinctive in animals that respond to anti-PD-1. These taxa may play a role in modulating the immune system and thus enhancing the effectiveness of immunotherapies. *Blautia* is known to produce acetate, a short-chain fatty acid that has been shown to contribute to the increase in the anti-tumor response of CD8^+^ T cells and decrease the risk of metastasis of breast cancer [78]. *Roseburia* and *Dorea* were found to be increased in the responders compared to the non-responders and are known butyrate protective taxa [79,80] These bacteria, as well as *Akkermansia*—not a biomarker but detected in the overall analyses barplot in the responder group—is known for its role in maintaining gut barrier integrity, which can prevent systemic inflammation and promote an environment conducive to effective immune surveillance and small frequencies colonize the oral cavity [79]. Also, *Akkermansia* produces propionate, which suppresses cell proliferation, migration, and invasion in colon cancer [81]. Interestingly, renal cell carcinoma and non-small cell lung carcinoma patients under PD-1 blockade showed a higher abundance of *Akkermansia,* which was also associated with IFN-γ release by T cells and prolonged progression-free survival [82,83,84]. This aligns with our results demonstrating higher T cell activation on PD-1 blockade responders. Together, these results suggest that the oral microbiota may influence immune responses and the outcome of PD-1 blockade therapy in mice with HPV^+^ oropharyngeal tumors.

We acknowledge that conventionally raised mice do not completely mirror humans’ physiology; however, they share at least 90% of the microbiome [85,86]. The human oral microbial composition comprises an abundance of *Streptococcus*, *Porphyromonas*, *Prevotella*, *Bifidobacteria*, and *Rothia* [24,87,88]. Nonetheless, the oral microbiota of mice has an abundance of Proteobacteria, *Pseudomonas*, *Staphylococcus*, and *Enterococcus* [89,90]. The differences in oral microbiota between mice and humans stems from various factors, including distinct microbial compositions and host interactions with these microbes, diet, and physiology. Despite these differences, both mice and humans experience dysbiosis in similar ways [91]. Importantly, our results demonstrate similarities in the microbiota composition between species, and we were able to identify relevant microbial species such as *Bacteroidetes* and *Blautia* previously identified in other human cancers and implicated in immunotherapy responses. This validates the utility of this animal model in identifying not only immune but also relevant microbial biomarkers for PD-1 blockade response in HNSCC.

While our study provides valuable insights, we acknowledge its limitations. First, our preclinical murine model was restricted to HPV16/18 oropharyngeal carcinoma. Second, we could not obtain sufficient amounts of DNA from swabs of the oral cavity, which limited our ability to characterize the microbiota at different time points. Lastly, for some analyses, we had a moderate to low sample size. Furthermore, we included molecules such as CTLA-4 and LAG-3, which are important for optimal regulation and homeostasis of T cells within the tumor microenvironment [92,93]. Given our focus on oropharyngeal carcinoma, it is important to include gamma delta T cells in our future analyses since these cells defend mucosa from pathogens and maintain homeostasis for further immunoregulatory functions [94]. We recognize that it is important to expand the scope of our research for other HPV subtypes, such as HPV-35, 33, and 39, for which, at the moment, there are no preclinical murine models but are highly prevalent in minority individuals. Therefore, longitudinal studies focused on the oral microbiota and immune phenotype in HNSCC patients, including those from minority backgrounds undergoing PD-1 blockade, are surely needed to confirm the utility of these microbial signatures as predictors of PD-1 blockade response [95].

## 5. Conclusions

In conclusion, there is a need to identify biological markers associated with PD-1 blockade response in HNSCC. Our study identified specific bacteria and immune phenotypes that could serve as biomarkers for immune checkpoint blockade response and may aid in identifying novel treatments for HNSCC patients.

## Figures and Tables

**Figure 1 cancers-16-02065-f001:**
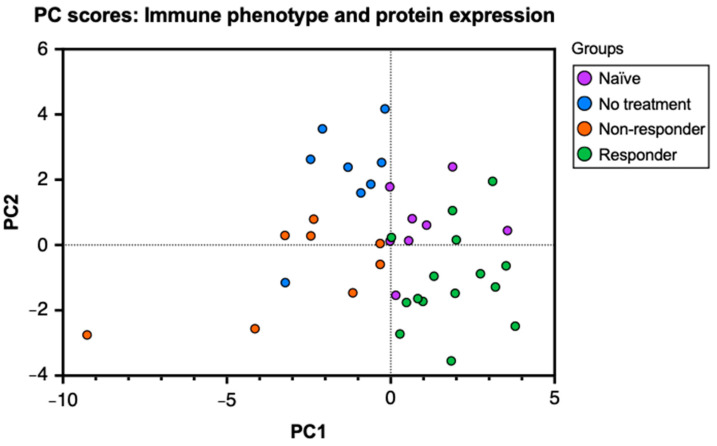
Principal component analysis (PCA) on cell type frequencies and protein markers differentiates anti-PD-1 therapy response. C57BL/6 mice were tongue-implanted with mEER tumor cells and treated with anti-PD-1 on days 10, 13, and 16. After 29 days of tumor implantation, tumor-draining lymph nodes, tongues, and tumors were collected from responders (tumor clearance), non-responders (no tumor clearance), tumor–implanted mice with no treatment, and a naïve group that did not have tumor implantation. PC scores plot of the principal component analysis (PCA) depicts the relationships between 26 markers for leukocyte frequencies and percent protein expression analyzed by flow cytometry from tumor-draining lymph nodes of the different mice groups.

**Figure 2 cancers-16-02065-f002:**
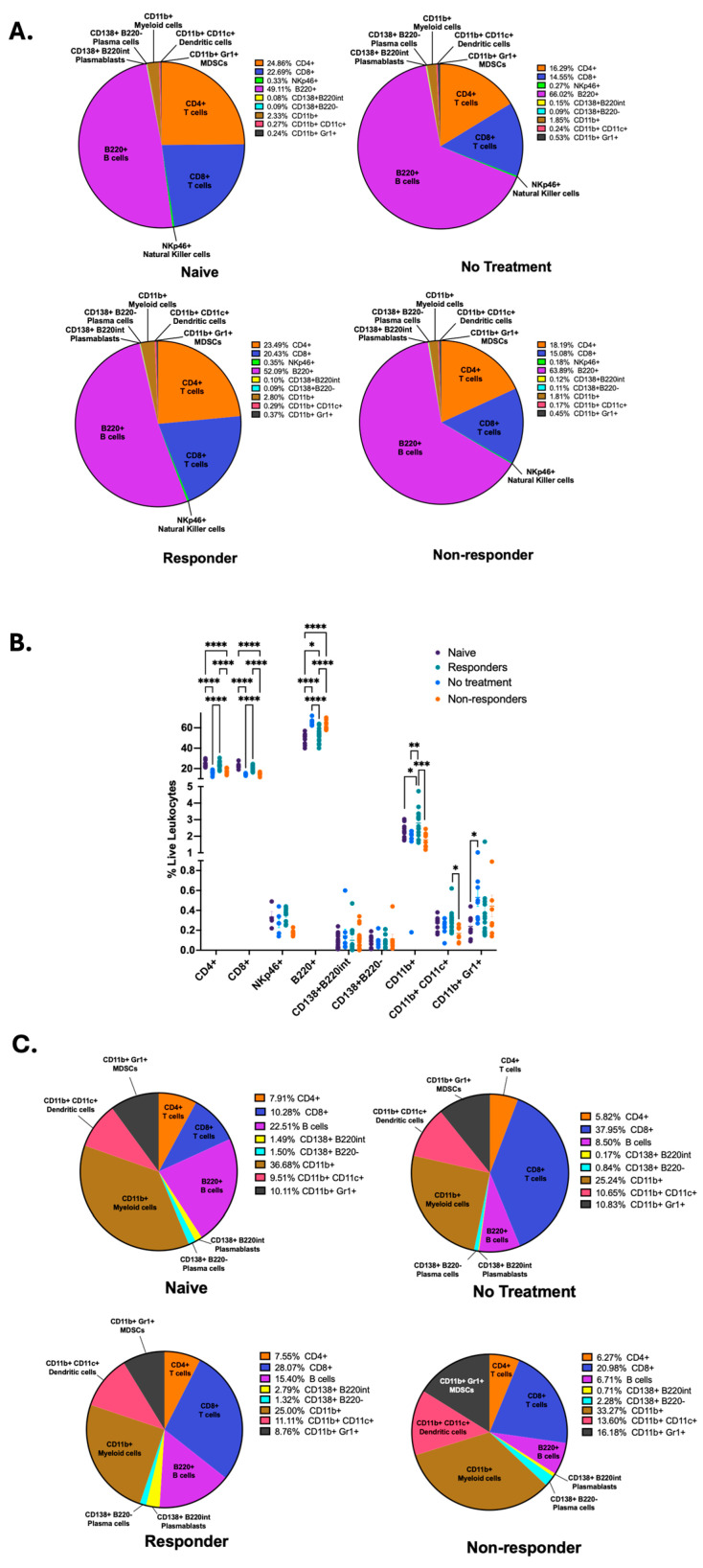
Anti-PD-1 therapy responders show increased frequencies of T cells, NK cells, myeloid, and dendritic cells in tumor-draining lymph nodes. Flow cytometry of tdLNs was performed to quantify the frequencies of CD4^+^, CD8^+^, NKp46^+^, B220^+^, CD138^+^B220int, CD138^+^B220^−^, CD11b^+^, CD11c^+^, and Gr1^+^ cells among live leukocytes and illustrated with parts-of-whole pie charts (**A**). The percent abundance of different leukocytes was quantified in tdLNs. A two-way ANOVA or a Kruskal–Wallis test was performed according to normality (**** *p* < 0.00001, *** *p* < 0.0001, ** *p* < 0.001, and * *p* < 0.01) (**B**). Flow cytometry immune cell frequencies from the tongue and primary tumor are illustrated with parts-of-whole pie charts (**C**). All data were pooled from two experiments (*n* = 41), except for NKp46 immune expression data (*n* = 25).

**Figure 3 cancers-16-02065-f003:**
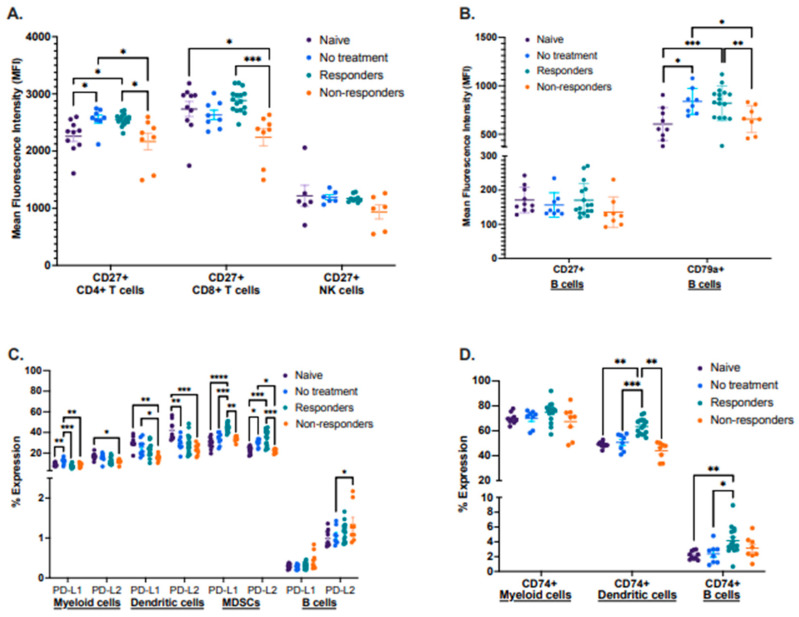
Differential expression of lymphocyte activation-associated proteins, inhibitory molecules, and antigen processing. Tumor-draining lymph nodes (tdLNs) were analyzed for the mean fluorescence intensity (MFI) of CD27 by CD4^+^ and CD8^+^ T cells as well as NK cells (**A**); the MFI of CD27 and CD79a by B cells (**B**); percent expression of PD-L1 and PD-L2 by myeloid, dendritic cells, MDSCs, and B cells (**C**), including percent expression of CD74 by myeloid, dendritic cells and B cells (**D**). Statistical analysis according to normality **** *p* < 0.00001, *** *p* < 0.0001, ** *p* < 0.001, and * *p* < 0.01.

**Figure 4 cancers-16-02065-f004:**
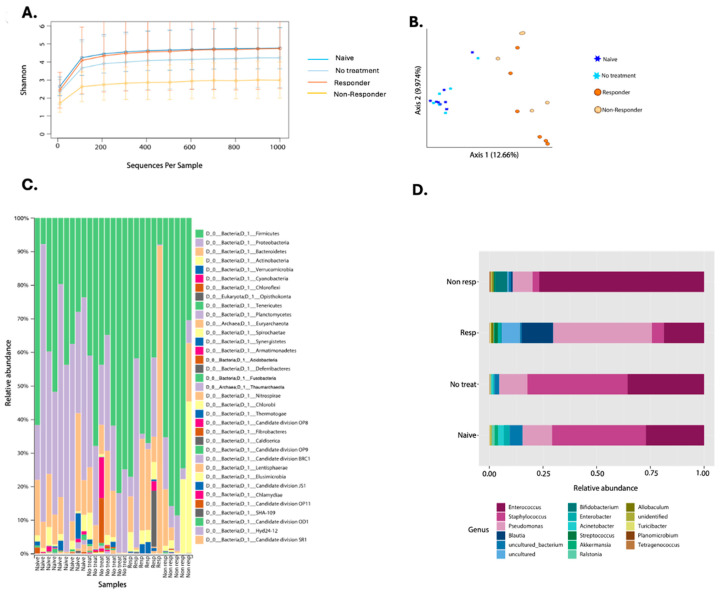
Composite figure showing diversity and taxonomy profiles according to anti-PD-1 therapy response in oropharyngeal carcinoma. Alpha diversity curves (Shannon Index) considering pooled data from two experiments on the oral microbiota of mice treated with anti-PD-1 (**A**). Beta diversity plot depicting Bray–Curtis distance is shown where stars represent the naïve (*n* = 9) and no treatment (*n* = 7) groups. In contrast, circles represent anti-PD-1 treated (responders *n* = 6 and non-responders (*n* = 5) (**B**). Taxonomic profiles of bacterial composition by anti-PD-1 treatment at phylum (**C**) and genus levels (**D**).

**Figure 5 cancers-16-02065-f005:**
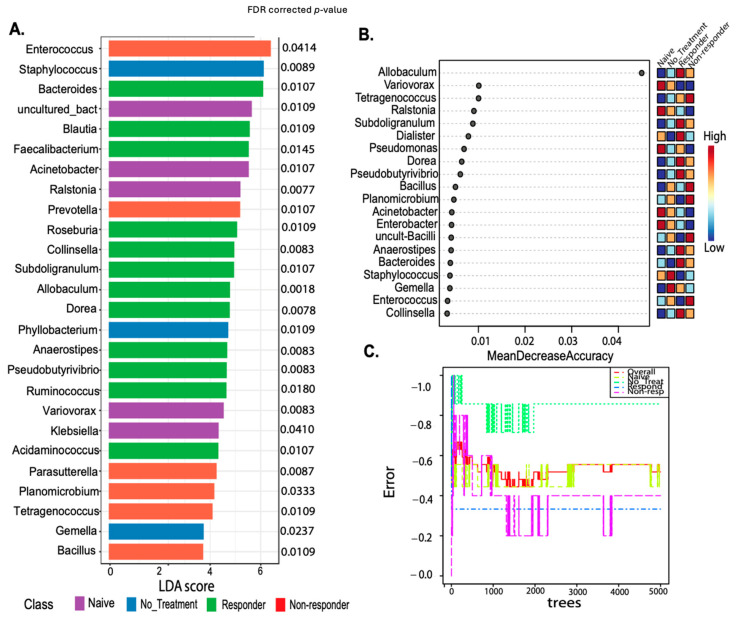
Putative biomarker analyses using both linear discriminant analysis effect size (LEfSe) and Random Forest comparing treatment response classes. LEfSE is shown as a barplot with significantly associated taxa with its FDR–corrected *p*-values (**A**). The Random Forest machine learning (ML) method shows a dot plot of the best taxa identified among a random subset of trees, which are determined by ranking the mean decrease accuracy (**B**). The Random Forest analyses show the cumulative error rates measured for each class. The overall error rate is shown as the red line, and other lines corresponding to each class are shown in the square legend. The OOB error is 0.519 (average error for each calculated using predictions from the trees). The class error for each group is naïve = 0.44, no_treatment = 0.857, responders = 0.333, and non-responders = 0.4 (**C**).

## Data Availability

The original data for the microbiota analysis presented in the study are openly available in QIITA under the study ID: 14411 and have been uploaded to the European Nucleotide Archive (ENA) with accession number PRJEB74643 ERP159289. The raw data supporting the conclusions of the flow cytometry data in this article will be made available by the authors upon request.

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
