# Peer review of "Immune and Microbial Signatures Associated with PD-1 Blockade Sensitivity in a Preclinical Model for HPV+ Oropharyngeal Cancer"

_cancers, 2024, doi:10.3390/cancers16112065_

Round 1

Reviewer 1 Report

Comments and Suggestions for Authors

Coments to author

The study by Jennifer focus on the biological marker identification associated to PD-1 blockade and define novel therapeutics for HNSCC patients. Despite the abundance of data in this study, the organization of the article is poor, making it very difficult to read. Additionally, there are already many similar literature reports on this topic, so the originality of this study is average. Therefore, considering these factors, it is recommended that the authors carefully refine the results before considering publication.

Major

The results of this study have been presented in a overly simplistic and rough manner. The authors should refine and organize the results carefully, categorize them for presentation, and ensure there is a logical flow between them to make the results comprehensible to the readers. The current version reads somewhat obscurely and lacks coherence. Moreover, when displaying the biomarkers of the disease in Figure 5, it is essential to include ROC curves rather than just bar charts.

Minor

Additionally, attention should be paid to certain notations, such as the need for superscript on the "+" in CD4+ T cells.

Comments on the Quality of English Language

Minor

Author Response

  1. We appreciate the comments and understand your concern about the originality of the study. Although several studies have analyzed the immune profiles and microbiota in HNSCC patients, to our knowledge, none have performed paralleled quantification of immune cells and markers associated to immune activation or inhibition in different tissues, as well as tumor microbiota. We now included a small study that we performed on blood to also identify the significance of the markers identified to potentially predict immune checkpoint blockade response. We have also explained in more detail the limitations of the study.
  2. We also acknowledge that the manuscript has different types of data that may be difficult to comprehend. We have taken your recommendation to categorize the sections to ease the reading of the manuscript, in addition to editing several parts of the manuscript.
  3. We thank the reviewer for the detailed comments. However, regarding Figure 5, the reviewer was unaware that these were not simple barplots but boxplots resulting from linear discriminant analysis effect size or LEfSe Biomarker Analyses. We have clarified this in methods, modified the figure, and added new complimentary analyses named random forest. Lefse is a biomarker discovery algorithm that employs a non-parametric factorial Kruskal-Wallis (KW) sum-rank test to identify differentially abundant taxa concerning a specific metadata category, followed by a linear discriminant analysis to determine the effect size of each identified taxa per group and rank them accordingly. This is now explained in more detail in the methods section, which was very brief in the previous version. Additionally, and instead of ROC curves which are normally not used for 16S amplicons, we used another complementary method to predict the taxa associated with each mice response group named Random Forest, which uses a model of supervised learning (decision trees) where the algorithm predicts the values using an if-else condition. It operates by constructing a multitude of independent decision trees (using bootstrapping) and predicting a specific taxon from all the trees (in this case, over 5000), and displaying the error rate to obtain the taxon identification. This is now thoroughly detailed in the methods section, and the figure now has 5 panels. We also modified the results to reflect this change.

  4. We have changed the + sign to superscript.

Reviewer 2 Report

Comments and Suggestions for Authors

This is a well designed and interesting study that seeks to parallel PD-1 therapy that is used for human OPCs in a mouse model. This model uses xenografts of mice with mouse HPV 16-E6/E7 expressions, and is previously reported ; Modic, Z., Cemazar, M., Markelc, B. et al. HPV-positive murine oral squamous cell carcinoma: development and characterization of a new mouse tumor model for immunological studies. J Transl Med 21, 376 (2023). https://doi.org/10.1186/s12967-023-04221-4. Carper MB, Troutman S, Wagner BL, Byrd KM, Selitsky SR, Parag-Sharma K, Henry EC, Li W, Parker JS, Montgomery SA, Cleveland JL, Williams SE, Kissil JL, Hayes DN, Amelio AL. An Immunocompetent Mouse Model of HPV16(+) Head and Neck Squamous Cell Carcinoma. Cell Rep. 2019 Nov 5;29(6):1660-1674.e7. doi: 10.1016/j.celrep.2019.10.005. PMID: 31693903; PMCID: PMC6870917.

However, previous studies do not provide the type of immunologic and microbiome  information as presented in this study. It is particularly important that this study evaluated PD-1 in relationship to antigen presentation which largely goes unrecognized as a product of cytotoxic inhibitory regulators and is a target for oral] microbiome.

A general negative to the study is  a lack of discussion regarding the presence and interaction of other inhibitory  cytotoxicity regulators .

In addition, oral mucosal immunity generates unique T cell/TCR repertoire (gamma/delta) emphasizing differences in antigen recognition and responses not identified in most non-oral tissues should be discussed.

Furthermore,  HPV associated oncogenesis is generally fixed upon a few subtypes but there is increasing awareness that non-16/18 subtypes are highly significant in their contribution to oncogenesis particularly in some segments of the population for example non-hispanic blacks. This model neglects this biology.

Moreover this study had the opportunity, but sadly did not address: metabolic and physiologic associations derived from the oral microbiome. For example, oral microbiome, particularly pathogens release metabolites and enzymes (e.g., endopeptidases) influencing specific essential and non-essential amino acid synthesis pathways. This relationship manifests as  DNA and RNA instability and a dampened tumor immune surveillance. This latter activity involves not only PD-1L/R but LGA-3, CTLA-4 and the release of immunosuppressive cytokines but results in CD4/CD8+ differentiation producing i T regulatory cells  and Th17 display. These associations require discussion to substantiate a high level of significance for this study.

The investigators acknowledge that their modeling in the mouse will associate with a microbiome that differs from human oral as to genera and species identified in humans. This point of possible determination of significance needs to be emphasized. 

Author Response

  1. We are grateful for your valuable feedback on our paper. We acknowledge the importance of studying other inhibitory cytotoxicity regulators, gamma/delta T cells, other HPV subtypes and microbial-derived metabolites and their association to immune checkpoint blockade and oropharyngeal development. We have edited the discussion section to discuss these topics and proposed the importance of future studies on these areas. Your insights will help us to improve our research in ongoing studies.
  2. We also edited the discussion paragraph that discusses the differences between mice and human microbiota to emphasize the importance of the study despite these differences.

Reviewer 3 Report

Comments and Suggestions for Authors

The manuscript: “Immune and microbial signatures associated with PD-1 blockade sensitivity in a preclinical model for HPV+ oropharyngeal cancer” describes the differences among mice bearing oropharyngeal tumors in response to anti-PD-1 therapy. The major caveat of the study is the interpretation of the obtained data. I have the following comments:

1.        The mEER model grows in male mice. Were the experimental animals individually caged? Stress developed due to the hierarchy in cages with male mice groups could involve the study results.

2.        According to the figures, the responders to the therapy markedly differed from untreated controls. Therefore, the changes in TME and oral microbiota composition seem to be a consequence of the treatment, not a prerequisite of the response. It seems that in responders, the treatment resulted in the infiltration/expansion of CD4+ and CD8+ T cells and DCs in the tumors and the changes in microbiota. Nevertheless, this does not explain, why the animals responded to the treatment. The data show that in responders the immune cells responded to the treatment, which is a well-known fact. Thus, the lines 411-414 do not make sense to me. How can these factors be used as biomarkers for PD-1 blockade response in HNSCC patients, if they occurred after the treatment? 

3.        In the case of microbiota, it seems that ICI application induced changes in the oral microbiota, as there are huge differences between the treated and untreated mice (Fig. 5). In most cases, these differences were much higher than those between responders and non-responders. Samples of oral microbiota taken at different time points would elucidate the kinetics of microbiota changes. At least 3 time points should be included – before the tumor implantation, before treatment, and at the end of the study.

Author Response

  1. We are grateful for your revisions and valuable feedback on our paper. We appreciate the comment regarding the relationship between hierarchy and immune responses. We found valuable information demonstrating that alpha male mice have an increase in metabolic demands as well as a shift in immunophenotypes favoring adaptative immunity compared to subordinate mice (Lee W, Milewski TM, Dwortz MF, Young RL, Gaudet AD, Fonken LK, Champagne FA, Curley JP. Distinct immune and transcriptomic profiles in dominant versus subordinate males in mouse social hierarchies. Brain Behav Immun. 2022 Jul;103:130-144. doi: 10.1016/j.bbi.2022.04.015. Epub 2022 Apr 18. PMID: 35447300). However, in this study, we did not focus on the hierarchy of mice and immune responses; we will consider this for our future studies. Also, separating the mice would induce additional stressors due to isolation which can also impact immune responses. Therefore, we maintain the animals in groups with a maximum of 5 per cage.
  2. Thank you for pointing out that the markers analyzed at the end of treatment may not be biomarkers, for this reason, we performed a whole blood analysis and identified PD-L1 on neutrophils and PD-L2 on B cells to be increased in non-responders before treatment as potential biomarkers. We also changed the title of Figure 3 on lines 397-398.
  3. We also agree that is important to elucidate the kinetics of microbiota changes to identify biomarkers before treatment, however, we were unable to obtain sufficient DNA from oral swabs to perform 16S rRNA sequencing. We address this limitation in the last paragraph of our discussion. We are also careful about the use of the word “biomarker” when describing our data.

Reviewer 4 Report

Comments and Suggestions for Authors

In recent years, many researchers have made efforts to identify diagnostic and prognostic factors in various cancers, while searching for the relationship between these biomarkers and the clinicopathological features of cancer.  Viruses, bacteria and other pathogens that make up the human body's microflora play a key role in various metabolic functions. However, changes in the composition of microflora can lead to the development of cancers, including oropharyngeal cancer. Moreover, many studies indicate, the microbiome plays an important role in oncogenesis

The article presented to me for review by Jennifer Diaz-Rivera et al. fits perfectly into this global research direction.

         The Authors tried to characterize immune profiles and the association of the oral microbiota in oropharyngeal tumors associated with anti-PD-1 treatment response in a preclinical murine model for HPV positive oropharyngeal cancer. Currently, surgery, chemotherapy, radiotherapy, and more recently, immune checkpoint inhibitors (ICIs) are used to treat OPSCC. One of the most common ICIs targets the lymphocyte immune inhibitory receptor PD-1. Unfortunately, this therapy has very poor therapeutic effectiveness OPSCC. Only in 20-30% of cases a positive response to therapy with full recovery is observed. 

         The research was carried out in an animal model - Male C57BL/6 mice and in cell lines - Mouse tonsil epithelial cells expressing HPV-16 E6 and E7 as well as H-Ras. The Authors evaluated Immunological profiles associated with PD-1 blockade response as well as changes in the microbiota associated with PD-1 blockade response. 

               These studies support the utility of this animal model to identify not only immunological but also relevant bacterial biomarkers for PD-1 blockade response in HNSCC.

Their results confirmed the identification of specific bacteria and immune phenotypes could serve as biomarkers for treatment response in HNC patients. 

The research performed was well documented in tables, figures and diagrams. 

88 references were correctly used both in the Introduction and in the Discussion.

However, I have a few comments/suggestions

1. Do the authors believe that the presented research has any limitations?

2. It seems necessary to confirm the obtained results in the human population 

Author Response

We are grateful for your revisions and valuable feedback on our paper. Our study certainly has limitations and we have detailed these limitations in the discussion section. We also acknowledge the importance of translating these results into humans and thus we proposed this as future studies in the discussion.

Round 2

Reviewer 1 Report

Comments and Suggestions for Authors

The author's revisions are insufficient to address the issues I raised. I maintain my opinion.

Comments on the Quality of English Language

Minor.

Reviewer 3 Report

Comments and Suggestions for Authors

The manuscript has been improved; I understand that the analysis of microbiota kinetics was not possible due to technical reasons. I have no other comments.